# Lipidomic Profiling Identifies Signatures of Poor Cardiovascular Health

**DOI:** 10.3390/metabo11110747

**Published:** 2021-10-29

**Authors:** Irma Magaly Rivas Serna, Michal Sitina, Gorazd B. Stokin, Jose R. Medina-Inojosa, Francisco Lopez-Jimenez, Juan P. Gonzalez-Rivas, Manlio Vinciguerra

**Affiliations:** 1International Clinical Research Center (ICRC), St Anne’s University Hospital, 53, 656 91 Brno, Czech Republic; rivasser@ualberta.ca (I.M.R.S.); michal.sitina@fnusa.cz (M.S.); gorazd.stokin@fnusa.cz (G.B.S.); juan.gonzalez@fnusa.cz (J.P.G.-R.); 2Division of Preventive Cardiology, Department of Cardiovascular Medicine, Mayo Clinic, Rochester, MN 55902, USA; medinainojosa.jose@mayo.edu (J.R.M.-I.); lopez@mayo.edu (F.L.-J.); 3Marriot Heart Disease Research Program, Rochester, MN 55902, USA; 4Department of Global Health and Population, Harvard T.H. Chan School of Public Health, Boston, MA 02115, USA

**Keywords:** cardiovascular health, lipidomics, sphingolipids, phospholipids, mass spectrometry

## Abstract

Ideal cardiovascular health (CVH) is defined for the presence of ideal behavioral and health metrics known to prevent cardiovascular disease (CVD). The association of circulatory phospho- and sphingo-lipids to primary reduction in cardiovascular risk is unclear. Our aim was to determine the association of CVH metrics with the circulating lipid profile of a population-based cohort. Serum sphingolipid and phospholipid species were extracted from 461 patients of the randomly selected prospective Kardiovize study based on Brno, Czech Republic. Lipids species were measured by a hyphenated mass spectrometry technique, and were associated with poor CVH scores, as defined by the American Heart Association. Phosphatidylcholine (PC), phosphatidylethanolamine (PE), lysophosphatidylcholine (LPC), lysophosphatidylethanolamine (LPE) species were significantly lower in ideal and intermediate scores of health dietary metric, blood pressure, total cholesterol and blood fasting glucose compared to poor scores. Current smokers presented higher levels of PC, PE and LPE individual species compared to non-smokers. Ceramide (Cer) d18:1/14:0 was altered in poor blood pressure, total cholesterol and fasting blood glucose metrics. Poor cardiovascular health metric is associated with a specific phospho- and sphingolipid pattern. Circulatory lipid profiling is a potential biomarker to refine cardiovascular health status in primary prevention strategies.

## 1. Introduction

Cardiovascular diseases (CVD) represent the most common cause of mortality and morbidity among adults with roughly 30% of deaths worldwide [1]. The most important behavioral risk factors of CVD are unhealthy diet, physical inactivity, and tobacco use. The effects of these behavioral risk factors may cause hypertension, dysglycemia, dyslipidemia and overweight/obesity. In 2010, the AHA suggested an ideal cardiovascular health (CVH) score to improve the cardiovascular health of the American population by 20% and to reduce deaths from CVD and stroke by 20% in 2020 [2]. The ideal CVH score reports the presence of seven ideal health parameters (non-smoking, body mass index <25 kg/m^2^, moderate and/or vigorous physical activity, a healthy diet consistent with current guideline recommendations, untreated total cholesterol <5.17 mmol/L, untreated fasting blood glucose <5.55 mmol/L and untreated blood pressure <120/<80 mm Hg) [2]. The Kardiovize study is a prospective cohort study to investigate the complex relationship of CVD and outcomes with a range of biological, psychosocial, environmental, behavioral, and economic factors in the urban population of Brno, the 2nd largest city (after the capital city, Prague) of Czech Republic, a landlocked country located in the heart of Europe [3,4,5,6,7,8,9,10,11,12,13]. The Kardiovize study evaluated ~1% of the population of Brno and revealed that only 19% of the people aged 25–64 years old have an ideal CVH score [14]. In addition to traditional lipid risk factors [low density lipoproteins (LDL) and very low density lipoproteins (LDL) cholesterol, triglycerides], alterations in the composition of other circulatory lipid classes—such as sphingolipids and phospholipids—may contribute to the pathogenesis of CVD, as a result of dietary habits, environmental factors or genetic background. Phospholipids and sphingolipids are mainly found in the cell membrane, and the main structural difference is that phospholipids consist of a glycerol backbone whereas sphingolipids consist of a sphingosine backbone. Abnormal metabolism of these lipid species has been related to inflammation, oxidative stress, endothelial erosion and atherosclerotic CVD [15,16,17]. Circulating lipids, including sphingolipids and phospholipids have been proposed to participate to the development of CVD: the trajectory of lipidomic abnormalities has been followed longitudinally in healthy subjects, or in patients who suffered minor or major cardiovascular events [18,19,20]. A complex relationship among individual sphingolipid and phospholipid species, anthropometric and metabolic parameters has been reported in an Australian cohort of 640 subjects with prediabetes [21]. In a Canadian cohort of 990 teenagers, few glycerophosphocholine (GPC) metabolites were associated with blood pressure, visceral fat, and fasting insulin [22]. Inclusion of plasma lipidomic profiles has been recently reported to improve upon traditional risk factors for the prediction of cardiovascular events in patients with diabetes enrolled in the American ADVANCE trial and the Italian Bruneck study [18,23]. Cardiovascular primary prevention is crucial for risk stratification and to prevent the insurgence of CVD, in subjects that did not suffer yet of CVD or of cardiovascular events [24,25]. There are no population-based cohort studies evaluating whether circulating phospholipid and sphingolipid profiles can discriminate between poor and intermediate/ideal CVH, in absence of CVD, as defined by the AHA [2]. In addition, there are no studies involving lipidomic analysis in Central Europe, where CVD is extremely prevalent, designed to prevent CVD. The objective of the present study is to determine the sphingo- and phospho-lipid profile [ceramide (Cer), sphingomyelin (SM), phosphatidylcholine (PC), phosphatidylethanolamine (PE), lysophosphatidylcholine (LPC), lysophosphatidylethanolamine (LPE)] using Liquid Chromatography Electrospray Ionization Tandem Mass Spectrometric (LC-ESI/MS) in the Kardiovize study associated with the cardiovascular health score and individual health parameters (BMI, smoking status, health score diet, physical activity, blood pressure, fasting blood glucose and cholesterol levels). The aim of this study was to identify lipid species that could outperform as biomarkers for cardiometabolic risk stratification in subjects without CVD.

## 2. Results

### 2.1. Study Population

The mean age was 54.3 ± 17.4 years old and 56.1% were men. The distribution of CVH metrics is reported in Appendix A. From the 461 subjects analyzed, the highest distribution corresponds to the ideal fasting blood glucose score with 78.7% of subjects, followed by the ideal smoking status (54.7%). The ideal and intermediate scores are around 40% for each score in physical activity, body mass index (BMI), blood pressure, and total cholesterol metrics. The healthy dietary metric is composed of intermediate scores (96.7%) and poor scored (3.3%). Overall, 203 participants reported poor CVH, 238 intermediate CVH, and only 20 participants ideal CVH. In this study population, approximately 108 individual lipid species were thus identified including 32 PC species, 23 PE species, 10 SM species, 17 LPC species, 15 LPE species, and 11 Cer species.

### 2.2. Comparison of Lipid Species across Categories of Behavioral CVH Metrics

Behavioral CVH metrics are composed of smoking, dietary metric, BMI, and physical activity, which were associated with each lipid class and species.

First lipid species across categories of behavioral CVH metrics were compared. In the smoking metric, 6 lipid species were selected at the level of FDR < 0.05 which varied between participants with poor, intermediate, or ideal CVH (Appendix A and Appendix A). Never and former smokers exhibited higher levels of LPE 22:6, LPE 16:0, LPE 20:5, PC 14:1_14:1, PE 18:2_16:1 and PE 18:0_18:0 than current smokers (Figure 1).

The comparison of lipid species between participants with poor or intermediate level of the health diet metric was reported in Figure 2. No participants reported an ideal healthy diet score. Participants with intermediate health diet score exhibited significantly lower levels of PC 20:0_20:1, PC 20:0_20:3, PE 18:0_18:2, LPE 22:6, LPE 18:0, PE 16:0_20:2, LPE 16:0, PC 18:0/18:0, PE 16:0_20:3, LPE 20:5, PE 18:2_20:2, PC 16:0_18:0, PE 18:1_22:6, PE 16:0_18:2, PC 16:0_20:2, LPC 20:1, LPE 18:2, and PC 18:0_20:0 than those with poor score (FCs are reported in Table 1). No significant difference in lipid concentrations was evident across categories of physical activity and BMI (Appendix A).

### 2.3. Comparison of Lipid Species across Categories of Clinical CVH Metrics

Lipid concentrations across categories of clinical CVH metrics (blood pressure, total cholesterol, and blood fasting glucose) were compared. In general, the ideal level for clinical CVH metrics was related to lower lipid concentrations. Concerning the blood pressure metric, 53 lipid species were identified that significantly varied across categories (Appendix A and Appendix A), with a decreasing gradient from poor to ideal level (Figure 3A). Similarly, 5 lipid species significantly differed across categories of the total cholesterol metric (Appendix A and Appendix A). Specifically, PC 18:0_20:5, PE 18:0_20:5, PC 16:0_20:5 and PE 18:1_22:6 decreased from poor to ideal level, while Cer d18:1/14:0 increased (Figure 3B). The increased concentration of Cer d18:1/14:0 was also observed in participants with blood fasting glucose metrics at the ideal level, while other 18 lipid species were under-represented when compared with participants at poor level (Figure 3C and Appendix A and Appendix A).

### 2.4. Effect of CVH Status on Lipid Profiles

Lipid concentrations were associated with participants with poor, intermediate, or ideal CVH status. Based on the ANOVA analysis 59 lipid species were selected with False Discovery Rate (FDR) <0.05 that varied across the categories (Appendix A and Appendix A). As reported for clinical CVH metrics, a decreasing trend of all lipid species from poor to ideal CVH status (Figure 4) was observed.

In the PLS-DA, using the CVH status (poor, intermediate, and ideal) as the response variable and lipid species as predictor variables. PLS-DA is a popular supervised algorithm for clustering or classification problems that aims to optimize separation between different groups of samples and to select discriminative variables. Accordingly, the score plot reported in Figure 5A showed that the two PLS-DA components explained 45.6% and 5.5% of the variability, respectively. In this plot, a partial separation of participants with poor, intermediate, and ideal CVH status was observed, which was confirmed by the permutation test (Appendix A). The VIP analysis also indicated the most important lipid species involved in this discrimination, including PE 18:0_20:5, PE 18:0_22:4, PE 16:0_16:1, PC 18:0_20:5, PC 16:0_20:5, PE 16:0_20:3, PC 16:0_16:1, LPE 20:3, SM d18:1/18:1, PC 18:0_20:3, PC 18:1_20:5, PE 18:1_22:4, PE 18:1_22:6, and PE 16:0_18:2 (Figure 5B).

The top-three lipid species (i.e., those with VIP >1.7) across categories of CVH status were compared. Concentrations of PE 16:0_16:1, PE 18:0_20:5 and PE 18:0_22:4 decreased from poor to ideal CVH status (Figure 6).

## 3. Discussion

In this study, the cardiovascular health parameters from an ideal to a poor score were associated with different concentrations of circulatory sphingolipids and phospholipids. Concretely, two main patterns were observed: (1) lower levels of PC, PE, LPC, and LPE individual species were found in ideal and intermediate scores of health dietary metric (for blood pressure, total cholesterol, and blood fasting glucose levels), compared to poor scores, and (2) on the contrary, higher levels of distinct PC, PE, and LPE individual species were found in never or former smokers compared to current smokers. Sphingolipids and phospholipids associated with behavioral CVH metrics. PC, PE, LPC, and LPE are the four lipid classes associated with smoking and dietary health CVH scores. Smoking behavior is associated mainly with lower levels of PE and LPE lipid species when compared to never smokers or quitters. Weir et al. evaluated non-smokers and smokers in a Mexican-American cohort: PE species (PE 32:1, PE 34:1, PE 34:3, PE 36:1, PE 36:4 and PE 36:5) and LPE species (LPE 18:1 and LPE 18:2) among others lipid classes were associated to smoking status [26]. A statistically significant correlation between smoking and increased total serum cholesterol, LDL and VLDL cholesterol, and triglyceride serum concentrations has been reported; this in turn participates to an increase of pro-atherogenic lipid oxidation [27,28]. Daily fat dietary intake contributes to modify the lipid profile. Plasma phospholipid fatty acids indicate the intake of fatty acids in the past few days [29]. Plasma saturated fat content is associated with dietary saturated fat but can be identified endogenously [30]; dietary intake is the main source of Omega-6 and Omega-3 PUFA [30]. In this study, lower levels of saturated lipid species (PC 18:0_18:0, PC 18:0_20:0, LPE 18:0, LPE 16:0), PC and PE, (PC 20:0_20:1, PE16:0_20:2, PC20:0_20:3) were found in participants with intermediate dietary health scores. High levels with species containing PUFA should be found in intermediary dietary scores that could indicate the presence of omega-3 and omega-6 lipid species consumed in the diet. However, ideal dietary health scores were not present in this population that could confirm this hypothesis. Sphingolipids and phospholipids associated with clinical CVH metrics. The blood pressure metric is associated with a higher number of individual lipids and classes than other clinical or behavioral scores. Fourteen PC species, 18 PE species, 2 LPC species, 8 LPE species, 10 SM species were found more abundant in poor blood pressure scores when compared to ideal blood pressure scores. In particular, the blood pressure metric was the only cardiovascular parameter associated with sphingomyelin species (SM d18:1/18:1, SM d18:1/14:0, SM d18:1/20:1, SM d18:1/16:0, SM d18:1/18:0, SM d18:1/20:0, SM d18:1/22:1, SM d18:1/24:1, SM d18:1/26:1, SM d18:1/24:0). The present findings are consistent with the literature. Shahin et al. reported that patients with hypertension consuming hydrochlorothiazide anti-hypertensive drug exhibited alterations of the sphingolipids including 4 SMs (N24:2, N24:3, N16:1, and N22:1) [31]; while Zheng et al. observed that higher SM levels were associated with a pattern of non-dipper hypertension [32]. Fernandez et al. also reported that SM38:2 was the only SM molecular species related to future cardiovascular events [33]. Sphingolipids, including ceramides and sphingosine 1-phosphate, have been associated with blood pressure homeostasis and vascular functions but not sphingomyelin [32]. However, HDL-sphingomyelin decreases after remarkable weight loss was associated with blood pressure improvement [33]. On the other hand, dyslipidemia linked to poor fasting blood glucose score is likely related to prediabetic or diabetic state, as demonstrated by recent lipidomic studies analyzing the lipidomic alterations in these patients [21,34,35]. In this present study, PE individual species are the main lipids modified within the poor fasting blood glucose metric category. Other studies have also found PE positively associated with the development of diabetes [35,36], corroborating the current findings. In general, high levels of total cholesterol are related to elevated LDL-cholesterol and apolipoprotein-B containing lipoproteins that transport cholesterol and other lipids throughout the body and initiate atherosclerosis [37]. Alterations of circulatory lipids in both poor fasting blood glucose and poor total cholesterol metric might be due to an increase in lipoprotein production and dysregulation of PC and PE containing omega-3 and omega-6 PUFA, which in turn are crucial for the synthesis of eicosanoids [34,38]. Moreover, adequate levels of lipid species containing PUFA and their biologically active derivatives have been shown to favor synaptic function and to prevent brain inflammation, providing a link between cardiometabolic and brain health [39]. Sphingolipids and phospholipids associated with CVH total scores. Across all categories of CVH metrics, PE species (PE 16:0_16:1, PE 18:0_20:5, and PE 18:0_22:4) are the most affected, decreasing significantly in poor versus ideal CVH status. A recent animal study reported that alterations in plasma lipid species might reveal metabolic health; plasma lipids are impacted by heritable factors despite dietary intake, and lipid species are modulated by loci spread across the genome [40]. In primary prevention, this study provides a lipidomic signature that can be used as a surrogate for cardiovascular health in absence of CVD events. Sphingolipid and phospholipid biosynthetic pathways associated to CVH. Among phospholipids, PC, LPC, PE, LPE, and sphingolipids (SM and Cer) play a major role in the pathogenesis of CVD [17,41]. Figure 7 indicates the lipid pathway of the lipids analysed in this present study.

PC, PE, LPC, LPE are structural and functional cell membrane components [42]. Approximately 90% of cardiac PC phospholipid is synthesized by the Cytidine Diphosphate (CDP)-choline pathway [43,44]. PE is synthesized by the CDP-ethanolamine pathway and the phosphatidylserine decarboxylase pathway. PE in turn is converted to PC by phosphatidylethanolamine N-methyltransferase (PEMT) [42]. PC biosynthesis is tightly associated with plasma lipoprotein homeostasis [45]. It is well known that PC in lipoproteins is converted to triglycerides in the liver [46,47]. Lysophospholipids (LPC and LPE) are deacylated products of phospholipids with one fatty acid chain and are produced by phospholipase A (PLA1 or PLA2) [48]. Production of LPC also results from lecithin-cholesterol acyltransferase activity or hepatic secretion [49]. Ceramides are complex sphingolipids biosynthesized in all tissues from saturated fat and sphingosine [50]. SM is the most abundant sphingolipid in lipoproteins and total plasma after PC [51]. The sphingomyelin-ceramide pathway has also been involved in the development of atherosclerosis since this pathway has a role in the oxidized LDL-induced of smooth muscle cells [52]. Interestingly, links between SM and ceramide levels were not identified. The ceramide d18:1/14:0 was the only ceramide species affected in this study. Elevated levels of circulating ceramides and ceramide ratios have been identified as risk factors for major cardiovascular events [53]. Nevertheless, the ceramide composed of d18:1/14:0 showed an opposite effect with higher values in ideal blood pressure, cholesterol, and glucose CVH scores. There are only a few studies indicating evidence of plasma Cer d18:1/14:0 as a risk or protective factor for CVD. For instance, Yao et al. reported that Cer d18:1/14:0 is a biomarker to identify the risk of acute coronary syndrome [54]. More studies are needed to understand the role of Cer d18:1/14:0. Comparisons with other studies and what does the current work add to the existing knowledge. Previous studies analysed the association between lipid species and CVD [19,55,56], or between lipid species and single metabolic risk factors [40,57]. This is the first study providing a sphingolipidomic and phospholipidomic signature that can be used as a fast surrogate for systemic cardiovascular health in absence of CVD events, currently requiring multiple biochemical and anthropometric assessments, if validated in independent cardiovascular cohorts. LC-ESI/MS platforms have become a widespread technology within clinical reference and referral laboratories world-wide, penetrating into hospitals and regional clinical laboratories. Therefore, a strength of this study relies a robust LC-ESI/MS platform to define in depth sphingo- and phospho- lipid profiles, combined with unique multidimensional cardiovascular health data, in a large size population-based cohort in Central Europe, a region where CVD incidence is the highest in the world. Limitations of the study include: (1) LC/MS method used provides information of the carbon chain length and number of bonds, but does not provide information of the position of the double bonds; (2) other phospholipids and plasmalogens were not quantified in this study, and triglycerides and diacylglycerides are species not retained in the column used; (3) participants with ideal health dietary metrics to cover all optimal categories were not found. The stringency of AHA criteria for CVH deserves careful consideration [58]; it is not possible from cross-sectional analyses to establish a causal link between altered lipidomic profiles and CVH.

## 4. Materials and Methods

### 4.1. Study Population and Data Collection

The Kardiovize study is a multidisciplinary cross-sectional epidemiological project [3,4,5,6,7,8,9,10,11,12,13,14], including 2160 participants aged 25–64, and 270 subjects aged ≥65. 65 years was thus used as cut-off point to define older adults. For this lipidomic analysis, 259 participants aged 25 to 64 years old (young cohort) were randomly selected, while 212 participants aged 65 to 89 years old that were selected to participate in this study were parents or foster parents. Therefore, older participants had a family relationship with younger participants. Ten participants aged 65 to 89 years old who presented a history of cardiac major events, such as stroke and coronary events, were excluded from this analysis. The present study includes 461 female and male participants aged 25–89 years old. Participants signed the informed consent and completed a health interview questionnaire, smoking history, medication intake, physical activity—using a validated Czech translation of the international physical activity questionnaire (IPAQL)—and a food frequency questionnaire (FFQ). Participants underwent a physical examination assessing blood pressure measurement using an automated office measurement device (BpTRU, model BPM 200; BpTRU Medical Devices Ltd., Vancouver, BC, Canada). Height and weight were measured using a digital medical scale with meter (SECA 799; SECA, GmbH and Co. KG, Hamburg, Germany). Laboratory tests were performed with 12-h fasting full blood samples using a Modular SWA P800 analyser (Roche, Basel, Switzerland) or enzymatic colorimetric methods (RocheDiagnostics GmbH, Hamburg, Germany), all assessed in St Anne’s University Hospital by trained research nurses. Serum blood samples were stored at −80 °C until lipidomic analysis was performed [59]. The study protocol was approved by the ethics committee of St Anne’s University Hospital, Brno, Czech Republic (reference 2 G/2012), in accordance with the Declaration of Helsinki.

### 4.2. Lipid Extraction and LC-ESI/MS Analysis

Serum lipids were extracted [60]. Briefly, 100 μL of serum containing deuterated lipid internal standard (Equisplash, Avanti Polar Lipids, Alabaster, AL, USA) was mixed with chloroform/methanol (2:1, *v*/*v*) and CaCl2 (0.025%, *w*/*v*) and shake with a thermoshaker (TS-100, Biosan, Riga, Latvia) at 1300 rpm. Samples were sonicated for 30 s (Kraintek 18, Hradec Kralove, Czech Republic). After centrifugation (16,000× *g*, 15 min, 4 °C; Eppendorf 5427R), the lower phase was extracted and the top layer was processed again following the same Folch extraction. Layers were combined and were evaporated using a speed vacuum. Lipid extracts were reconstituted with acetonitrile:water (3:1, *v*/*v*). Lipid extractions were analyzed using a HPLC system composed of a Thermo Scientific Dionex UltiMate^TM^ 3000 RSLCnano system connected to a ABSciex QTRAP 6500 system, as previously described [61]. A Waters Acquity UPLC BEH HILIC, 1.7 μm, 2.1 × 100 mm (Waters, Milford, MA, USA) column with a guard column, Acquity UPLC BEH HILIC 1.7 μm VanGuard Pre-Column 2.1 × 5 mm (Waters, Milford, MA, USA). The column was eluted a flow rate of 400 μL/min a linear gradient from 0 to 10 min to 80% mobile phase A (water/acetonitrile [5:95, *v*/*v*] with 10 mM ammonium acetate) and over 11 min to 98% mobile phase B (water/acetonitrile [50:50, *v*/*v*] with 10 mM ammonium acetate; pH = 8.2). Mass spectrometry analyses were performed using electrospray ionization in the positive/negative mode and MRM scan, with a resolution of 0.7 +/− 0.1 u [Full Width at Half Maximum (FWHM)]. MS additional parameters were as follows: curtain gas (CUR) 35; temperature (TEM) 500 °C; Ion Source Gas 1 (GS1) 40; Ion Source Gas 2 (GS2) 50; Ion spray voltage 5200/−4500. A library of theoretical precursor ions was generated for Cer, SM, PC, PE, LPE, and LPC varying the length of ceramide or fatty acid carbon chain. The characterization of phospholipid and sphingolipid classes and individual species was achieved in positive mode (Cer and SM) with the formation of [M+H]^+^, in negative mode (PE, LPE) with the formation of [M-H]^−^ and [M+OAc]^−^ for LPC and PC. The collision energy varies from 43 to 51 eV. Data analysis was carried out using the Skyline daily 4.2.1.19058 software (MacCoss Lab, Department of Genome Sciences, University of Washington, Seattle, WA, USA).

### 4.3. Statistical Analyses

Lipidomic analyses were performed on the MetaboAnalyst web tool (www.metaboanalyst.ca (accessed on 27 October 2021)). Prior to further analysis, missing values were replaced using the k-nearest neighbors (KNN) algorithm, and data were normalized using their median, log-transformed e auto-scaled. The concentration of lipid species was compared for each CVH metric, and across categories of CVH status. The volcano plot—which combines fold-change analysis and *t*-test—was generated for two-group comparisons. Lipid features with fold-change threshold >2 or <0.5 and *p*-value < 0.05 were considered statistically significant. The one-way Analysis of Variance (ANOVA) followed by Fisher’s least significant difference (LSD) was applied for multigroup comparisons. Lipid features with adjusted *p*-value < 0.05 using the False Discovery Rate (FDR) were considered statistically significant. In general, the average concentrations of important lipid features identified were compared and reported using a heatmap. The Partial Least Squares Discriminant Analysis (PLS-DA) was performed to predict the variability across categories of CVH status. The Variables Important in Projection (VIP) analysis was used to identify important lipid features that contributed to the separation between categories.

## Figures and Tables

**Figure 1 metabolites-11-00747-f001:**
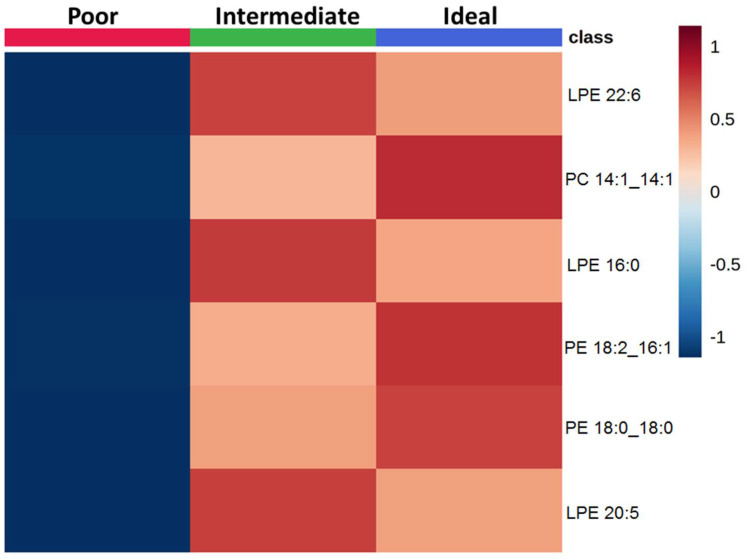
Comparison of lipid species across categories of smoking status. The heatmap shows the average concentrations of important lipid features identified by the ANOVA analysis and grouped by categories of smoking status.

**Figure 2 metabolites-11-00747-f002:**
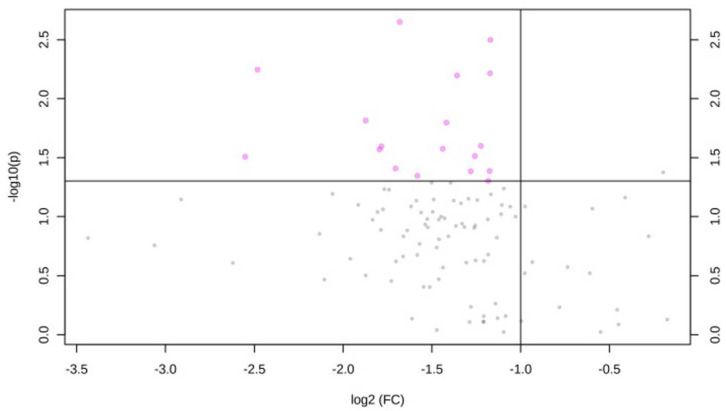
Comparison of lipid species across categories of diet. Volcano plot of important lipid features identified by the comparison between poor and intermediate dietary categories. This plot shows the magnitude (log 2 Fold-change of intermediate versus poor category; *x*-axis) and significance (−log 10 *p*-value; *y*-axis) for the comparison of lipids between intermediate and poor scores of dietary CVH metric. Lipid species with fold-change >2 or <0.5 and *p*-value < 0.05 are represented as pink dots.

**Figure 3 metabolites-11-00747-f003:**
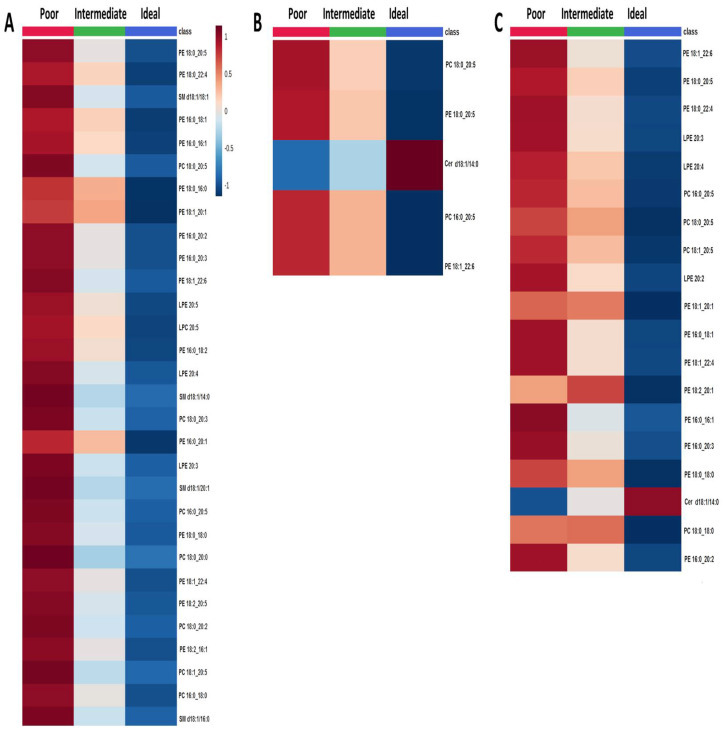
Comparison of lipid species across categories of 564 (**A**) blood pressure, (**B**) total cholesterol, and (**C**) fasting blood glucose metrics. The heatmap shows the average concentrations of important lipid features identified by the ANOVA analysis and grouped by categories of each metric.

**Figure 4 metabolites-11-00747-f004:**
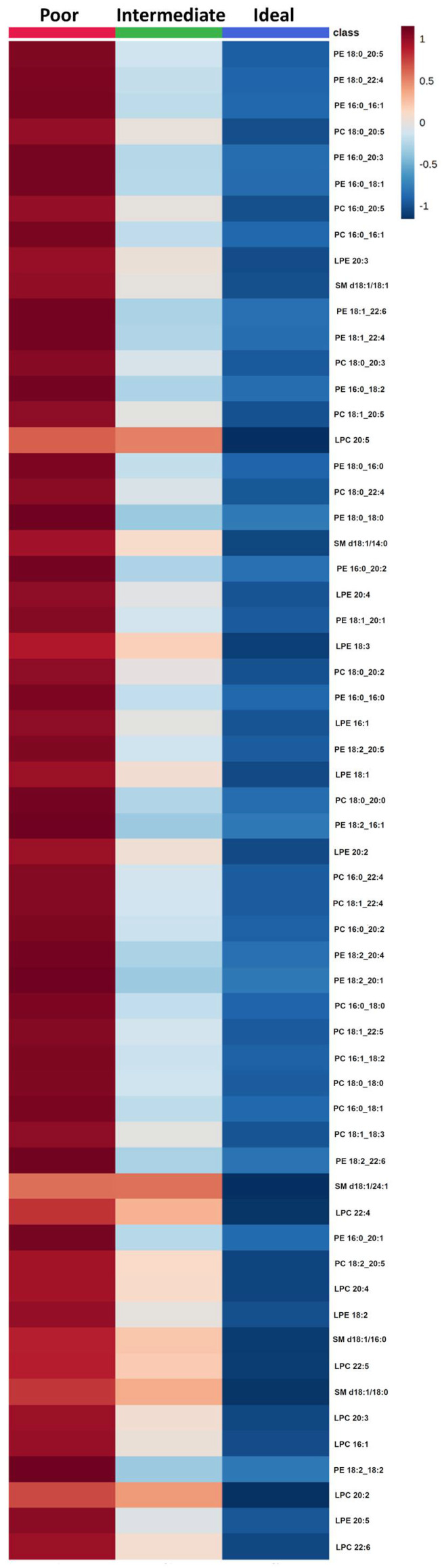
Comparison of lipid species across categories of cardiovascular health. The heatmap shows the average concentrations of important lipid features identified by the ANOVA analysis and grouped by categories of cardiovascular health.

**Figure 5 metabolites-11-00747-f005:**
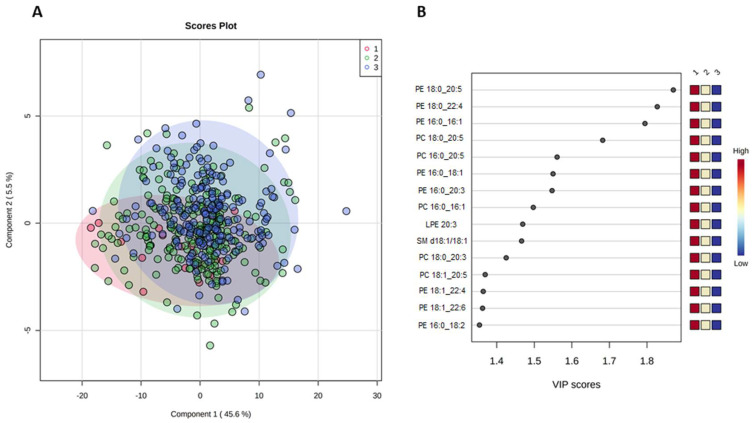
Outputs of the PLS-DA of lipid species on the cardiovascular health status. (**A**) PLS-DA scoreplot and (**B**) importance of variables ranked by Variable Importance in Projection (VIP) score for the comparison between categories of cardiovascular health. 1 = Poor; 2 = intermediate; 3 = ideal.

**Figure 6 metabolites-11-00747-f006:**
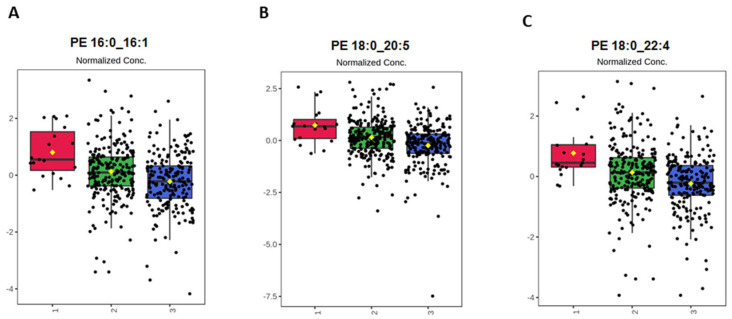
The top-three discriminant lipid species (**A**) PE 16:0_16:1, (**B**) PE 18:0_20:5, and (**C**) PE 18:0_22:4 across categories of CVH status. The boxplots show lipid species with VIP >1.7.

**Figure 7 metabolites-11-00747-f007:**
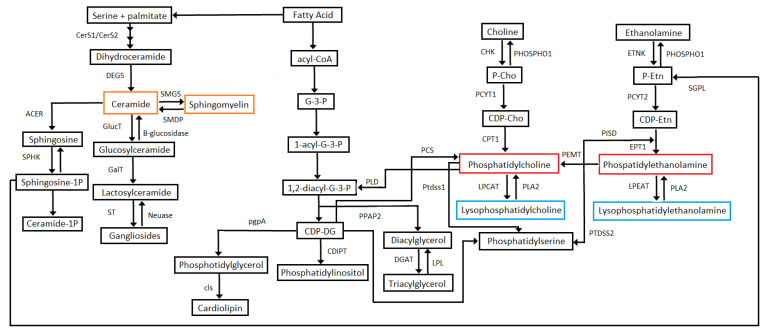
Schematic Representation of lipid metabolic pathways. Metabolite abbreviations: G-3-P, sn-Glycerol 3-Phosphate; 1-acyl-G-3-P, 1-Acyl-sn-glycerol 3-phosphate; 1,2-diacyl-G-3-P, 1,2-Diacyl-sn-glycerol 3-phosphate; CDP-DG, CDP-1,2-diacylglycerol; P-Cho, Choline Phosphate; CDP-Cho, CDP-choline; P-Etn, Phosphoethanolamine; CDP-Etn, CDP-ethanolamine. Enzyme abbreviations: CerS1/CerS2, Ceramide Synthase 1/Ceramide Synthase 2; DEGS, sphingolipid delta-4 desaturase; SMGS, sphingomyelin synthase; SMDP, Sphingomyelin Phosphodiesterase; ACER, alkaline ceramidase; SPHK, sphingosine kinase; GlucT, Glucosylceramide-Transferase; GalT, Galactosylceramide-Transferase; ST, Sialyl-Transferase; Neusase, Neuraminase; pgpA, phosphatidylglycerophosphatase A; cls, cardiolipin synthase; PLD, Phospholipase D1/2; PPAP2, phosphatidate phosphatase; CDIPT, CDP-diacylglycerol-inositol 3-phosphatidyltransferase; CHK, choline kinase; PHOSPHO1, phosphoethanolamine/phosphocholine 586 phosphatase; PCYT1, choline-phosphate cytidylyltransferase; CPT1, diacylglycerol cholinephosphotransferase; PCS, phosphatidylcholine synthase; DGAT, diacylglycerol O-acyltransferase; LPC, lipoprotein lipase; Ptdss1, phosphatidylserine synthase 1; LPCAT, lysophosphatidylcholine 650 acyltransferase; PLA2, secretory phospholipase A2; ETNK, Ethanolamine Kinase; PCYT2, ethanolamine-phosphate cytidylyltransferase; PISD, phosphatidylserine decarboxylase; EPT1, ethanolaminephosphotransferase; PEMT, phosphatidylethanolamine N-methyltransferase; LPEAT, lysophospholipid acyltransferase; PTDSS2, phosphatidylserine synthase 2. Lipid species analysed in the present study: Orange rectangles = sphingolipids, red rectangles = phospholipids, blue rectangles = lysophospholipids.

**Table 1 metabolites-11-00747-t001:** Important lipid features selected, fold changes (FCs) and *p*-values.

Lipid Species	Fold-Change	*p*-Value
PC 20.0/20.1	0.31	0.002
PC 20.0/20.3	0.45	0.003
PE 18.0/18.2	0.18	0.006
LPE 22.6	0.44	0.006
LPE 18.0	0.39	0.006
PE 16.0/20.2	0.27	0.015
LPE 16.0	0.37	0.016
PC 18.0/18.0	0.43	0.025
PE 16.0/20.3	0.29	0.025
LPE 20.5	0.37	0.027
PE 18.2/20.2	0.29	0.027
PC 16.0/18.0	0.42	0.031
PE 18.1/22.6	0.17	0.031
PE 16.0/18.2	0.31	0.039
PC 160/202	0.44	0.041
LPC 20.1	0.42	0.041
LPE 18.2	0.33	0.045
PC 18.0/20.0	0.44	0.049

## Data Availability

The datasets used and/or analysed during the current study are available from the corresponding author on reasonable request.

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
