# Peer review of "Lipidomic Profiling Identifies Signatures of Poor Cardiovascular Health"

_metabolites, 2021, doi:10.3390/metabo11110747_

Round 1

Reviewer 1 Report

  1. The resolution of the mass spec analysis shopuld be cited in the methods.
  2. Since plasma GPL containing polyunsatured fatty acids at 2n-2 supply these essential fatty acids to the brain. In the discussion it would be important to discuss the negative effects of too low levels of GPL on brain function. It may well be a delicate balance between vascular and brain health in relation to these lipids.

Author Response

Responses to Reviewer 1

  1. The resolution of the mass spec analysis shopuld be cited in the methods.

The resolution of the mass spec analysis has been added to the methods.

  1. Since plasma GPL containing polyunsatured fatty acids at 2n-2 supply these essential fatty acids to the brain. In the discussion it would be important to discuss the negative effects of too low levels of GPL on brain function. It may well be a delicate balance between vascular and brain health in relation to these lipids.

We followed this recommendation, and discussed briefly the effects of too low levels of GPL on brain function.

Reviewer 2 Report

The study by Rivas Serna and colleagues investigates the association of CVH metrics with the circulating lipid profile, serum sphingolipid and phospholipid species, in a population-based cohort. Towards this, lipids species were measured by a hyphenated mass spectrometry technique, and were associated with poor CVH scores. In particular, Phosphatidylcholine (PC), phosphatidylethanolamine (PE), lysophosphatidylcholine (LPC), lysophosphatidylethanolamine (LPE) species were significantly lower in ideal and intermediate scores of health dietary metric, blood pressure, total cholesterol and blood fasting glucose compared to poor scores. This is an interesting study, with potential application in future CVD preventive strategies.

Comments to the authors:

1) From the 461 individuals analyzed how many were represented in each category of CVH status (poor, intermediate or low)? This information does not appear to be included anywhere in the manuscript and is unclear to me.

2) In Fig 2A the information provided in the volcano plot is rather difficult to see. Could the authors provide a higher resolution image, potentially in higher magnification? Also could they include clarifications regarding the information shown in the plot eg pink vs grey samples shown on the plot?

3) In consistency with the remaining figures, maybe Figure 2B could also be presented as heat map. This would help conveying the observed differences in a very clear manner. Could the authors clarify in the figure legend if this shows only intermediate health diet? Also, what about the other health diet individuals? Could these be included in the figure too?

4) In page 7, line 150, could the authors briefly explain what PLS-DA is? It is only mentioned in the methods section, which is at the end of the manuscript.

5) Based on their findings, what do the authors believe is the optimal circulatory lipid profiling (lipidomic signature) that could potentially be used as biomarker for CVD? Maybe a schematic diagram could help towards this. Along the same lines, how do they envision that their findings would help in future CVD preventive strategies?

Author Response

Responses to Reviewer 2

The study by Rivas Serna and colleagues investigates the association of CVH metrics with the circulating lipid profile, serum sphingolipid and phospholipid species, in a population-based cohort. Towards this, lipids species were measured by a hyphenated mass spectrometry technique, and were associated with poor CVH scores. In particular, Phosphatidylcholine (PC), phosphatidylethanolamine (PE), lysophosphatidylcholine (LPC), lysophosphatidylethanolamine (LPE) species were significantly lower in ideal and intermediate scores of health dietary metric, blood pressure, total cholesterol and blood fasting glucose compared to poor scores. This is an interesting study, with potential application in future CVD preventive strategies.

We thank the Reviewer for his/her comments.

Comments to the authors:

1) From the 461 individuals analyzed how many were represented in each category of CVH status (poor, intermediate or low)? This information does not appear to be included anywhere in the manuscript and is unclear to me.

We apologize for the missing information that has been added in the revised version.

2) In Fig 2A the information provided in the volcano plot is rather difficult to see. Could the authors provide a higher resolution image, potentially in higher magnification? Also could they include clarifications regarding the information shown in the plot eg pink vs grey samples shown on the plot?

As requested, we have improved the resolution of Figure 2 and we have clarified the meaning of pink dots in the figure legend.

3) In consistency with the remaining figures, maybe Figure 2B could also be presented as heat map. This would help conveying the observed differences in a very clear manner. Could the authors clarify in the figure legend if this shows only intermediate health diet? Also, what about the other health diet individuals? Could these be included in the figure too?

As reported in the paragraph “Study population”, the healthy dietary metric is composed of intermediate scores (96.7%) and poor scores (3.3%). For this reason, these findings are reported through the Volcano plot. In particular, results are reported as the fold change in the intermediate group if compared to the poor category. We have clarified this point in the figure legend.

4) In page 7, line 150, could the authors briefly explain what PLS-DA is? It is only mentioned in the methods section, which is at the end of the manuscript.

As requested, we have briefly explained what is PLS-DA in the paragraph “Effect of CVH status on lipid profiles”

Reviewer 3 Report

This is an interesting study where the authors aimed to search for an association between AHA Cardiovascular Health (CVH) score (and its individual CVH metrics) and a lipidomic profile in a cohort of 461 individuals randomly selected from Kardiovize study. By using LC-ESI/MS, the authors determine sphingolipid and phospholipid species (Cer, SM, PC, PE, LPC, LPE) in serum. The study is well-designed and performed according to the hypothesis, and the results are interesting. However, there are some specific comments to improve the manuscript.

Minor points:

  1. In line 78, there are abbreviations (Cer, SM, PC, PE, LPC, LPE) which are not previously defined. The abstract is an independent part of the manuscript.
  2. This reviewer does not understand the following numbers found in the manuscript, concretely in lines:
  • 200: In particular, the blood pressure metric was the only “284” cardiovascular parameter associated with sphingomyelin species
  • 261: Ceramides are complex sphingolipids biosynthesized in “327” all tissues from saturated fat and sphingosine
  • 284: LC/MS method used provides information of the carbon chain length and number of bonds, but does not provide information “349” of the position of the double bonds

Major points:

  1. The authors might consider including a descriptive table of the population. It would be more informative for the readers.
  2. For the lipidomic analysis, the authors selected 259 participants aged 25 to 64 years old, and 212 participants aged 65 to 89 years. As the authors indicate, 65 years old was used as a cut-off to define elder people. So, why did the authors perform the analysis considering of the participants together? Why did not performed the study separating by age according the mentioned cut-off?
  3. Were there differences by gender? Did the authors consider performing the study separating the participants by gender?
  4. The discussion turns out to be extremely long and slightly daunting, due to the need of justify the number of results obtained. However, the conclusion at the end of the discussion is vague. Is the lipidomic signature a valid surrogate marker to define CVH? What is the additional value of the lipidomic signature to the CVH defining parameters or CVH score itself? Which is the concrete lipidomic signature that determines CVH status?

Author Response

This is an interesting study where the authors aimed to search for an association between AHA Cardiovascular Health (CVH) score (and its individual CVH metrics) and a lipidomic profile in a cohort of 461 individuals randomly selected from Kardiovize study. By using LC-ESI/MS, the authors determine sphingolipid and phospholipid species (Cer, SM, PC, PE, LPC, LPE) in serum. The study is well-designed and performed according to the hypothesis, and the results are interesting. However, there are some specific comments to improve the manuscript.

-We thank the Reviewer for his/her words of appreciation.

Minor points:

In line 78, there are abbreviations (Cer, SM, PC, PE, LPC, LPE) which are not previously defined. The abstract is an independent part of the manuscript.

-Thank you. We have defined the abbreviations where it was indicated.

This reviewer does not understand the following numbers found in the manuscript, concretely in lines:

200: In particular, the blood pressure metric was the only “284” cardiovascular parameter associated with sphingomyelin species

261: Ceramides are complex sphingolipids biosynthesized in “327” all tissues from saturated fat and sphingosine

284: LC/MS method used provides information of the carbon chain length and number of bonds, but does not provide information “349” of the position of the double bonds

-We apologize for these mistakes. These numbers belonged to a previous formatting of the manuscript, and they should not be there. The numbers mentioned by the reviewer were now deleted.

Major points:

The authors might consider including a descriptive table of the population. It would be more informative for the readers.

-We thank for the Reviewer for this suggestion. The distribution of the 7 CVH metrics (BMI, smoking status, health score diet, physical activity, blood pressure, fasting blood glucose and cholesterol levels) in the study population is already reported in Table S1, while we have indicated that the mean age was 54.3±17.4 years old and 56.1% were men in the first sentence of the Results section. As such, we propose that an additional table would not add valuable information.

For the lipidomic analysis, the authors selected 259 participants aged 25 to 64 years old, and 212 participants aged 65 to 89 years. As the authors indicate, 65 years old was used as a cut-off to define elder people. So, why did the authors perform the analysis considering of the participants together? Why did not performed the study separating by age according the mentioned cut-off?

-In the Kardiovize study, the young and the old cohorts were designed and recruited as described, and 65 years old was used as a cut-off to define elder people, according to the literature and to WHO and to CDC guidelines. However, the baseline information about age that we provided in the Methods did not represent a rationale for the design of the present study. Age is a major and well-known risk factor for CVD, and analyses comparing young versus old would not meaningful in our opinion, or would not add value to the manuscript. In fact, the American Heart Association (AHA) does not include age among the individual health parameters (BMI, smoking status, health score diet, physical activity, blood pressure, fasting blood glucose and cholesterol levels) characterizing the ideal cardiovascular health (CVH). Our aim was to identify a possible lipid signature for ideal CVH applicable to the general population, irrespective of age cut-offs.

Were there differences by gender? Did the authors consider performing the study separating the participants by gender?

-There were no differences by gender, hence we considered opportune not to describe the study with separation of the participants by gender difference.

The discussion turns out to be extremely long and slightly daunting, due to the need of justify the number of results obtained. However, the conclusion at the end of the discussion is vague. Is the lipidomic signature a valid surrogate marker to define CVH? What is the additional value of the lipidomic signature to the CVH defining parameters or CVH score itself? Which is the concrete lipidomic signature that determines CVH status?

-Thank you for these comments. We attempted to formulate clearer answers in the revised Discussion section, as it follows:

Is the lipidomic signature a valid surrogate marker to define CVH? What is the additional value of the lipidomic signature to the CVH defining parameters or CVH score itself?

-“This is the first study providing a sphingolipidomic and phospholipidomic signature that can be used as a fast surrogate for systemic cardiovascular health in absence of CVD events, currently requiring multiple biochemical and anthropometric assessments, if validated in independent cardiovascular cohorts. LC-ESI/MS platforms have become a widespread technology within clinical reference and referral laboratories worldwide, penetrating into hospitals and regional clinical laboratories. Therefore, a strength of this study relies a robust LC-ESI/MS platform to define in depth sphingo- and phospho- lipid profiles, combined with unique multidimensional cardiovascular health data, in a large size population-based cohort in Central Europe, a region where CVD incidence is the highest in the world.”

Which is the concrete lipidomic signature that determines CVH status?

-We propose the following start of the Discussion: “In this study, the cardiovascular health parameters from an ideal to a poor score were associated with different concentrations of circulatory sphingolipids and phospholipids. Concretely, two main patterns were observed: 1) lower levels of PC, PE, LPC, and LPE individual species were found in ideal and intermediate scores of health dietary metric (for blood pressure, total cholesterol, and blood fasting glucose levels), compared to poor scores, and 2) on the contrary, higher levels of different PC, PE, and LPE individual species were found in never or former smokers compared to current smokers.”

The individual species of PC, PE, LPC, and LPE categorized per CVH metrics were subsequently described in the text, in the frame of the current literature.

Round 2

Reviewer 3 Report

The authors addressed all the issues raised by this reviewer.